# Trends and Changes in Socio-Economic Inequality in Self-Rated Health among Migrants and Non-Migrants: Repeated Cross-Sectional Analysis of National Survey Data in Germany, 1995–2017

**DOI:** 10.3390/ijerph19148304

**Published:** 2022-07-07

**Authors:** Elisa Wulkotte, Kayvan Bozorgmehr

**Affiliations:** 1Department of Population Medicine and Health Services Research, School of Public Health, Bielefeld University, P.O. Box 10 01 31, 33501 Bielefeld, Germany; wulkottee@rki.de; 2Section of Health Equity Studies & Migration, Department of General Practice & Health Services Research, Heidelberg University Hospital, Im Neuenheimer Feld 130.3, 69120 Heidelberg, Germany

**Keywords:** inequality, equity, migration, social disparity, trends, slope index of inequality, relative index of inequality, socio-economic status, refugees, social epidemiology

## Abstract

Socio-economic inequalities in health may change over time, and monitoring such change is relevant to inform adequate policy responses. We aimed to quantify socio-economic inequalities in health among people with direct, indirect and without migration background in Germany and to assess temporal trends and changes between 1995 and 2017. Using nationally representative survey data from the Socio-Economic Panel (SOEP), we quantified absolute and relative socio-economic inequalities in self-reported general health by calculating the slope (SII) and relative index of inequality (RII) with 95% confidence intervals (CI) among each group and year (1995–2017) in a repeated cross-sectional design. Temporal trends were assessed using a GLM regression over the SII and RII, respectively. The total sample size comprised 492,489 observations, including 108,842 (22.23%) among people with migration background. About 31% of the population with and 15% of the population without migration background had a low socio-economic status. Socio-economic inequalities in health persisted in the group with migration background (1995 to 2017), while inequalities in the non-migrant population increased (SII: $\beta_{Trend}$ = 0.04, *p* < 0.01) and were on a higher level. The highest socio-economic inequalities in health were found among those with direct migration background ($\beta_{SII,\;min}$ = −0.23, *p*< 0.01; $\beta_{SII,\;max}$ = −0.33, *p* < 0.01). The results show that the magnitude and temporal dynamics of inequalities differ among populations with direct, indirect and without migration background. Monitoring systems can capture and investigate these inequalities if migrant populations are adequately integrated into the respective systems.

## 1. Introduction

Socio-economic inequalities in health are dynamic and amenable to change over time with respect to their magnitude and the patterns of inequalities presented across different societal groups [1]. As such, contemporary research on health inequalities is less concerned with the question of whether or not socio-economic inequalities exist, but rather with the challenge of adequately monitoring social inequalities [2] to promote and inform adequate policy responses aimed at reducing socio-economic inequalities in health [3]. Several initiatives at global and national levels have been proposed to improve health inequality monitoring [2,4] in order to align with the Sustainable Development Goals of leaving no one behind. However, socio-economic inequalities among migrant populations, i.e., the concurrent monitoring of both vertical dimensions of inequality with horizontal dimensions such as migration, have barely been considered in these debates. Health information systems, even in countries with strong economies, are not only challenged to generate health data stratified by socio-economic status over time [4,5]. They often fail, at the same time, to provide health data at a population level stratified by adequate indicators of migration, such as country of birth, nationality or residence status [6]. Therefore, socio-economic inequalities in health among migrants, and changes over time in such inequalities, remain understudied.

In Germany, few studies have been conducted on changes in socio-economic inequalities in health over time [7], and those that exist focus only on the general population [7,8,9,10,11,12]. While the studies show that socio-economic inequalities in health have persisted [7,10,11,13] or increased [7,8,12] over time in the general population, the magnitude and the temporal dynamics of socio-economic inequalities in health among migrants remains unclear. We hence aimed to quantify the magnitude of socio-economic inequalities in health among people with direct and indirect migration background living in Germany and to assess temporal changes in inequalities *vis-á-vis* the magnitude of and change in socio-economic inequality in the country’s population without migration background.

## 2. Materials and Methods

### 2.1. Data Source

We used nationally representative data from the German Socio-economic Panel (SOEP) between 1995 and 2017. The SOEP is a longitudinal panel study, which enrols annually more than 20,000 individuals from about 10,000 households. Data on a wide range of individual, demographic, social and economic aspects, including some questions on self-reported health outcomes, are collected via structured personal interviews [14].

In addition, there are subgroup-specific surveys that have been added and expanded since the first survey in 1984. These include, among others, an immigration sample that was added in 1995 and the IAB-SOEP-migration sample conducted by the Institute for Employment Research (IAB) under the SOEP at the German Institute for Economic Research (DIW) Berlin. The first wave in 2013 contains information from almost 5,000 individuals with migration background, going beyond the previously queried topics and offering more migration-specific information [15]. Five further waves followed until 2018. In addition, the IAB-BAMF-SOEP contains a survey of refugees, which was first conducted in 2016 and has been repeated annually since then. This is performed jointly by the IAB, the SOEP at DIW Berlin and the Research Center Migration, Integration and Asylum of the Federal Office for Migration and Refugees (BAMF-FZ). In the first survey, 4,527 refugees who arrived in Germany between 2013 and 2016 were interviewed. The survey includes specific questions about flight and the asylum procedure [16].

### 2.2. Design, Sampling and Participants

In the current study, we used data from 28 waves of SOEP surveys within a repeated cross-sectional analysis between 1995 and 2017. The data comprise 1,121,763 person-years and 131,767 individuals from 123,037 households. We excluded the so-called “high income” SOEP-sample, a sample that consists of extraordinary top-income individuals from Germany, from our analysis to avoid an overrepresentation of the high-income groups in our study population, which can lead to biases with regard to the socio-economic and income-specific characteristics examined. After further exclusion of individuals younger than 18 years, the sample comprised 849,190 person-years and 128,981 individuals from 41,264 households.

We included participants based on information on their migration background, defined as not having German citizenship by birth or having at least one parent who does not have German citizenship by birth [17]. This led to a categorisation of the study population into participants without migration background (i.e., German citizenship by birth and both parents having German citizenship; non-migrant group) and with migration background (migrant group).

As the definition of migration background yields very heterogenous social categories that are likely to be characterised by large *within*-group variability with respect to their socio-economic position, we further specified the population with migration background to reflect a potential history of immigration as well as refugee migration through the asylum system: direct migration background (individuals who immigrated themselves), indirect migration background (individuals whose mother, father or both immigrated but who were born in Germany) and direct migration background with refugee history.

### 2.3. Health Outcome

We used self-rated general health (SRH) as a dependent variable. The variable was captured in SOEP using a one-item question asking, “How would you describe your current state of health?” The response options ranged from “very good” (1) to “bad” (5) and were dichotomized for the analyses, resulting in a binary variable with the expressions “good health” (response options “very good”, “good”) and “poor health” (response options “satisfactory”, “poor”, “bad”) [18,19].

### 2.4. Socio-Economic Status

We operationalised socio-economic status (SES) using three indicators of socio-economic position (education, occupation, income) that were transformed to an index according to Lampert et al. (2013) [20].

Education was measured according to the “Comparative Analyses of Social Mobility in Industrial Nations” (CASMIN) classification, which distinguishes nine educational categories [21]. Based on the scoring approach described in Lampert et al. (2013), the nine categories were transformed to three groups for descriptive purposes. The indicator for occupational status was measured according to the “International Socio Economic Index of Occupational Status” (ISEI) [22], which in turn builds on the “International Standard Classification of Occupations” (ISCO-88) combined with information on education and income to derive occupational status groups [21]. In contrast to the approach in Lampert et al. (2013), we used occupational status as an individual measure and applied a scoring that considers people without employment in the scoring system. For descriptive analyses, occupational status was classified as an ordinal variable with four categories: (1) without employment, (2) low ISEI score (score range: 16–40), (3) moderate ISEI score (score range: 41–65), (4) high ISEI score (score range: 66–90).

The income indicator was monthly net equivalent household income, which is based on the net household income variable recorded in the SOEP and was calculated following the Organization for Economic Co-operation and Development (OECD) equivalence scale [23].

The values of these three indicators were each given points from 1 to 7. Based on the sum of these three scales, the SES index was constructed as metric variable (range: 3–21). Following Lampert et al. (2013), the metric score was transformed to quintiles (Q1–Q5), wherein SES was considered as low (Q1), moderate (Q2–Q4) and high (Q5).

### 2.5. Analysis

Analysis was conducted with Stata Version 16.0.

We performed a descriptive analysis of the absolute and relative frequencies of health outcomes, socio-economic variables and SES stratified by sex, age groups (18–39 years, 40–65 years, over 65 years) and migration indicators for each year and for the total observation period.

We then quantified socio-economic inequality in self-rated health by calculating the slope (SII) and relative index of inequality (RII). SII and RII are measures of absolute and relative inequality and are appropriate for measuring changes in inequality over time, as they consider the size of the underlying population used to quantify inequalities and are hence robust against temporal variations in the size of socio-economic groups that may otherwise affect the magnitude of inequalities. The SII indicates the absolute difference in health between the group with the lowest SES and the group with the highest SES, taking into account the health status of all groups in between and the underlying population sizes of each group. The RII can be interpreted according to a difference in relative risk, i.e., the relative difference in health status between the lowest and the highest SES groups [24]. These analyses were performed for the years 1995 to 2017. Inequalities among people with indirect or direct migration background and asylum and refugee history could only be analysed for the years 2000 to 2017, as the sample sizes in the previous years were too small in the group with asylum and refugee history, and comparability could not be ensured.

For the calculation, the socio-economic characteristics (independent variable) were individually divided into deciles for each survey year. Each decile was assigned a cumulative rank value between zero (lowest SES) and one (highest SES). This measure is called a “ridit-score”. The health variable SRH was dichotomized so that the expressions could be coded as zero (characteristic of interest is not present) and one (characteristic of interest is present). The dependent variable was a poor level of subjectively reported general health status. The calculations of SII and RII were carried out using a generalized linear model (GLM), as proposed in the literature [25,26] and already conducted by a large number of researchers [11,13,27,28]. For the calculation of SII and RII with SRH as the dependent variable, a binomial distribution was assumed. The two indices SII and RII were calculated using the GLM as follows:$$g\left(Y\right)=\beta_{0}+\beta_{1}ridit+\epsilon$$

The constant in the equation is β_0_, while β_1_ represents the coefficient, which were analysed as SII and RII, respectively. In SRH, the target event *Y* = 1 was poor health, while good health was coded as *Y* = 0. Based on the coding chosen, a negative correlation indicated that people with low SES were more likely to have poor subjective health than people with high SES. That is, if the SII was negative or the RII was less than 1, the risk of poor SRH was reduced by the corresponding difference from 0 or 1 from the lowest to the highest SES group. For example, with an SII of −0.15, the absolute risk difference is 0.15, which is 15 percentage points and is pronounced to the disadvantage of people with low SES. For example, if the RII value is 0.90, the relative difference in health is 0.10 between the lowest and highest SES groups, to the disadvantage of low SES people. This means that people with low SES have a 1.10-fold increased chance of having a poor SRH.

Calculations were adjusted for sex and age, insofar as they had not been stratified. Stepwise regressions were applied using the best model following the Akaike Information Criterion (AIC) and Bayesian Information Criterion (BIC). To ensure comparability, the same model with the same variables was chosen for the strata to be compared in each year. Thus, in the analyses for the groups with and without migration background, the characteristics age and sex were controlled for, but not in the analyses for the groups with indirect and direct migration background or with asylum and refugee history.

Time-trend analysis was conducted by calculating GLM for each stratum. For this purpose, an interaction term was generated with the “ridit-score” variable and a new time variable. To generate the time variable, the variable observation year, which included the years from 1995 to 2017, was recoded according to approaches applied in the literature [11,13,27,28] and had 23 values between zero (1995) and one (2017). The intervals between the respective values were equal. The regression coefficient of the interaction term indicated whether there was a linear trend. If it was greater than zero (SII) or one (RII), a linear trend in the increase could be assumed. If the regression coefficient was zero, there were no changes, and if it was less than zero (SII) or one (RII), this indicated a trend of decrease. This increase and decrease were not related to the differences, but to the values of the regression coefficients. This means that, with RII values of, for example, less than or equal to one and a trend value less than or equal to one, this can be interpreted as an increase in the differences.

## 3. Results

### 3.1. Descriptive Results

A total of 849,190 observations (46.80% male individuals) were considered for the analysis during the observation period (Table 1). Of these, 77.77% had no migration background. Among those with migration background (108,842 observations), 62.21% had a direct and 21.72% an indirect migration background, while 14.32% immigrated as asylum seekers or refugees (Table 1). About 31% of the group with migration background and 15% of those without migration background had a low SES, while a high SES was found for 10% of the population with and 20% of the group without migration background.

Throughout the survey years 1995, 2000, 2005, 2010 and 2015, the proportion of individuals with direct migration background decreased, while the proportion of those with indirect and refugee background increased (Table 1). It should be mentioned here that the study population of people with asylum and refugee histories in 2016 and 2017 was significantly larger than in previous years, with up to 6088 (Appendix A). Compared to 1995, the proportion of individuals with low SES in the sample in 2015 was halved, while the proportion of those with high SES was more than doubled. The proportion of individuals in poor vs. good SRH persisted over time (Table 1). Detailed descriptive data of the study population for all survey years are reported in the Supplementary Material, including missing values for each variable described (Appendix A: Descriptions of the study population by sex, age, migration status, SES, educational status, income, occupational status and SRH, 1995–2017).

### 3.2. Socio-Economic Inequalities in SRH of People with and without Migration Background

In all years, significant absolute (SII) and relative (RII) socio-economic inequalities in SRH could be demonstrated for people with and without migration background. Overall, the inequalities within the group without migration background were higher than within the group with migration background. Figure 1 and Figure 2 show the trends over time. The difference by SII was lowest for people without migration background in 1995 and 1997, with −0.18 (*p* < 0.01), and highest in 2016 ($\beta_{SII}$ = −0.32; *p* < 0.01). For SII of people with migration background, no trend can be seen initially. While the inequality increased from 2009 to 2010, a decrease was observed from 2011 onwards. In 2015, the SII was −0.23 (*p* < 0.01). The values for 2016 ($\beta_{SII}$ = −0.09, *p* < 0.01) and 2017 ($\beta_{SII}$ = −0.08, *p* < 0.01) deviated strongly from the trend of previous years (Figure 1). This discrepancy can be seen in absolute and relative inequalities, regardless of sex and age (see Appendix A). The relative health inequalities were similar to the absolute ones. The RII varied between 0.83 (*p* < 0.01) and 0.55 (*p* < 0.01) for people with migration background and between 0.78 (*p* < 0.01) and 0.59 (*p* < 0.01) for people without migration background (Figure 2). The trend analyses showed a slight increase in absolute and relative inequality for people without a migration background (SII: $\beta_{Trend}$ = −0.04, *p* < 0.01; RII: $\beta_{Trend}$ = 0.96, *p* < 0.01). For people with migration background, a slightly decreasing trend was indicated for the absolute inequality ($\beta_{Trend}$ = 0.04, *p* < 0.05) and an increasing trend for the relative inequality ($\beta_{Trend}$ = 0.92, *p* < 0.01).

A subgroup analysis by gender showed that the absolute inequalities (SII) among people without migration background were larger in almost all years than for people with migration background, regardless of sex. Absolute and relative inequalities in SRH in men and women with and without migration background are reported as graphs in the Appendix A. In the group without migration background, these inequalities were at a higher level for women than for men in almost all years. This remains true for women and men with migration background for most of the years. In the group with migration background, differences in SII between men and women could be seen from 1995 to 2004 with no trend detectable. In both sexes, the absolute inequalities (SII) increased from 2005 to 2011 up to −0.26 (*p* < 0.01). The relative inequalities were comparable, and the values ranged from 0.86 (*p* < 0.05) to 0.53 (*p* < 0.01) for women and from 0.82 (*p* < 0.01) to 0.55 (*p* < 0.01) for men with migration background. Linear trends in the group with migration background could be identified for relative inequalities in terms of an increase for women ($\beta_{Trend}$ = 0.89, *p* < 0.05) and men ($\beta_{Trend}$ = 0.67, *p* < 0.01), and for absolute inequalities, an increasing trend could be identified only for men ($\beta_{Trend}$ = −0.20, *p* < 0.01). In the group without migration background, linear trends indicating an increase in SII could be identified for both sexes (men: $\beta_{Trend}$ = −0.21, *p* < 0.01; women: $\beta_{Trend}$ = −0.03, *p* < 0.05), while a declining trend in RII could only be demonstrated for males ($\beta_{Trend}$ = 0.69, *p* < 0.01).

When comparing the SII and RII of the age groups 18 to 39 years old, 40 to 65 years old and over 65 years old, it can be clearly seen that the development as well as the magnitude of the inequalities varied. Figures of SII and RII can be found in the Appendix A. While for people without migration background aged 18 to 39 and over 65, the SII was between −0.12 (*p* < 0.01) and −0.29 (*p* < 0.01) and between −0.10 (*p* < 0.05) and −0.24 (*p* < 0.01), respectively, the values for the middle age group were higher and varied between −0.23 (*p* < 0.01) and −0.43 (*p* < 0.01). In the group without migration background, significant inequalities could be proven in all age groups for the whole observation period. This held for the middle age group with migration background. In the age group 65 years and older, some values were not significant, and in the group 18 to 39 years with migration background, nearly half of the values showed no statistical significance. For people with migration background, SII values were lowest in the youngest age group. In the age group 40 to 65 years, absolute inequalities increased from 2005 ($\beta_{SII}$ = −0.14, *p* < 0.01) to 2012 ($\beta_{SII}$ = −0.41, *p* < 0.01) and decreased from then on. A similar development was seen for the oldest age group, with an SII of −0.41 (*p* < 0.01) in 2012. The developments of relative inequalities were very similar to the ones of absolute inequalities, both for people with and without migration background. In the group with migration background, the significant values varied between 0.54 (*p* < 0.01) and 0.73 (*p* < 0.05) in the age group 18 to 39 years, between 0.53 (*p* < 0.01) and 0.80 (*p* < 0.01) in the age group 40 to 65 years and between 0.66 (*p* < 0.01) and 0.79 (*p* < 0.05) for people over 65 years. For those without a migration background, linear trends demonstrating an increase in relative and absolute inequalities were found regardless of age group. In contrast, for those with migration background, the trend analyses did not reveal significant values in any age group.

### 3.3. Socio-Economic Inequalities in SRH of People with Indirect and Direct Migration Background as Well as with Asylum and Refugee Background

Figure 3 and Figure 4 show the absolute (SII) and relative (RII) socio-economic inequality of people with indirect and direct migration background. Within the group of asylum seekers and refugees, no significant inequality based on SII and RII was identified. For people with an indirect migration background, a period with significant values of SII and RII was identified from 2010 to 2017. In these years, a decline of the SII was recorded from −0.21 (*p* < 0.01) in 2011 to −0.11 (*p* < 0.05) in 2016. Only among people with direct migration background could health inequality be demonstrated for the entire observation period. The SII varied between −0.23 (*p* < 0.01) and −0.33 (*p* < 0.01) for people with direct migration background and showed a slightly increase from 2005 to 2017 (Figure 3). The inequalities through the RII ran similarly to those of the SII. The trend analyses only revealed a significant value in relation to the SII for the group with an indirect migration background, which indicates that absolute socio-economic inequality in the SRH remained constant over time ($\beta_{Trend}$ = −0.00, *p* < 0.05).

## 4. Discussion

The analysis shows that socio-economic inequalities in SRH among people with migration background persisted, while inequalities among the non-migrant population in Germany increased. With regard to the age groups, it is noticeable that the inequalities in the group of 18- to 39-year-olds were clearly lower than those in the other age groups. It is also striking that the absolute and relative inequalities among people without migration background were higher than among people with migration background, regardless of age and sex. An exception is the group of people over 65, where the absolute and relative inequality among people with a migration background was much higher. Particularly relevant are inequalities among people with direct migration background, as these were considerably larger than those of all other groups studied across all years in both absolute and relative terms. These patterns seem to reflect the importance of subgroup analyses and the consideration of the heterogeneity of population groups. Addressing attempts to reduce avoidable inequalities and increase health equity, policy makers and practitioners should pay particular attention to people with direct migration background, as well as older people with migration background.

Notably, according to the examined SOEP data, socio-economic inequalities in SRH among people with asylum and refugee background did not exist. This stands in stark contrast to other survey studies that use measures of *subjective* social status to measure socio-economic inequalities in health among refugees [29,30], as opposed to the classical “objective” indicators of socio-economic positions captured by SOEP and used in this study to construct the SES index. A reason for this could be that the conventional indicators of socio-economic position (education, occupation, income) do not sufficiently capture the social status of asylum seekers and refugees, as the asylum system “evens out” or levels any differences in social status by an equal amount of welfare transfers and by imposing restrictions on access to job markets so that educational differences do not translate into differences in socio-economic position and health. As such, the results should not be interpreted in a sense that no social inequalities exist among the heterogenous groups of asylum seeker and refugees, but rather show that conventional approaches of measuring SES in migrant populations [31] have limitations that need to be overcome to adequately monitor socio-economic inequalities among refugee populations. Including subjective measures of social status and the self-rated changes in social status attributed to the migration process [30] can be a promising way forward and deserves further consideration for inclusion in national surveys on a routine basis.

Overall, the study demonstrates the potential of routine survey systems in place that capture both vertical dimensions of inequality, i.e., socio-economic variables, and relevant variables that allow stratification of the population into meaningful migrant groups. As a consequence of the results shown, a stronger engagement seems to be required to reduce inequalities among people with direct migration background, e.g., through adequate social policies that aim at early integration into labour, educational and social systems, in order to avoid the socio-economic differences that translate into large health gaps. Furthermore, interventions are needed that acknowledge migrants as a heterogenous population group and focus on improving the access to health care of those with a lower SES to overcome the socio-economic inequities in health. However, to be fully relevant to health monitoring among migrants, more health and health care indicators are required within the SOEP data.

This first study of temporal changes in socio-economic inequalities among migrant populations in Germany has several strengths and weaknesses. Overall, we used the best available data on a national level that is comparable over time and space to quantify inequalities and their temporal changes. We used the SII/RII as appropriate measures for the quantification of inequalities overtime and hence considered both absolute and relative dimensions of inequality in line with international recommendations for inequality monitoring [24,32]. The weaknesses relate to strata with small sample sizes as a result of the stratification procedure and the limitations of the subjective information as the data did not derive from objective measured information, especially with regard to the SRH and the missing possibility of statements about causality because of the cross-sectional analysis design used in this study. Concerning the variables SRH and SES, weaknesses refer to missing information due to combined outcomes. This study follows Lampert et al. (2013) in generating the SES. Thus, a valid and established instrument was used. However, there were some deviations from Lampert et al. (2013), and the newly calculated SES was not tested for construct validity.

## 5. Conclusions

This study shows that socio-economic inequalities in migrant populations in Germany constantly existed from 1995 to 2017 and differed between migrant groups. Monitoring systems offer the opportunity to capture and investigate these inequalities and highlight groups among which inequalities persist or rise, as this study shows using SOEP-data. The precondition for this is the recording of required characteristics, as well as a large sample size that allows for subgroup analyses. With growing migrant populations, not only in Germany, it will become increasingly important to understand the living situations of heterogenous migration populations and the circumstances in which social and economic policies interact with their social wellbeing. This includes monitoring the socio-economic inequalities in health in order to overcome them in the sense of the sustainable development goals. This aim can only be achieved if migrant populations are adequately integrated in monitoring systems.

## Figures and Tables

**Figure 1 ijerph-19-08304-f001:**
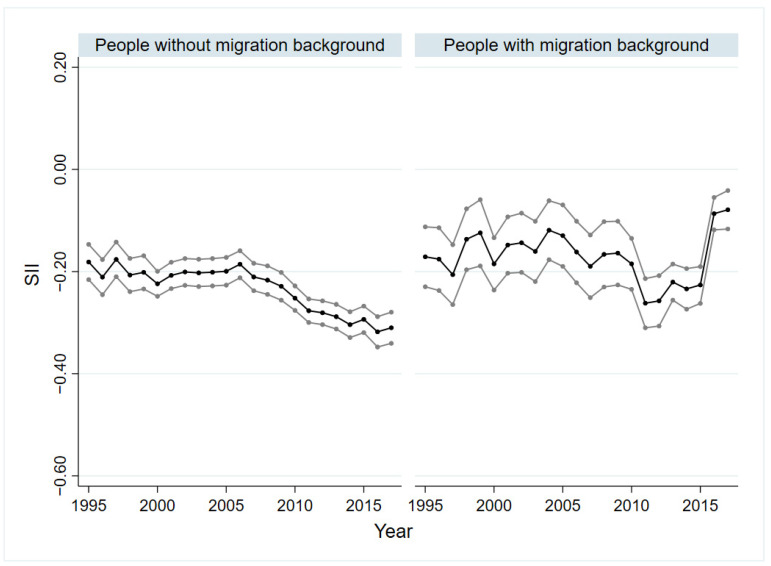
Absolute socio-economic inequalities in SRH in people with and without migration background, 1995–2017. SRH: self-rated health; SII: slope index of inequality; black line: regression coefficient; grey line: 95% confidence interval.

**Figure 2 ijerph-19-08304-f002:**
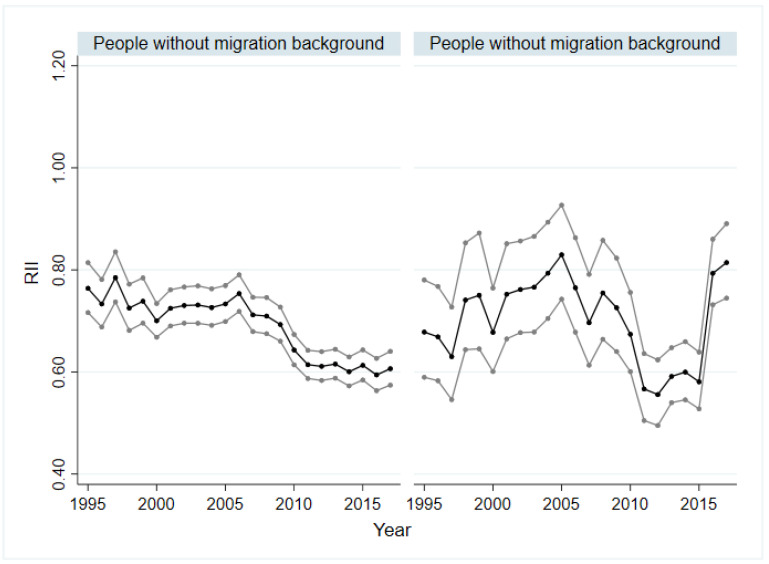
Relative socio-economic inequalities in SRH in people with and without migration background, 1995–2017. SRH: self-rated health; RII: relative index of inequality; black line: regression coefficient; grey line: 95% confidence interval.

**Figure 3 ijerph-19-08304-f003:**
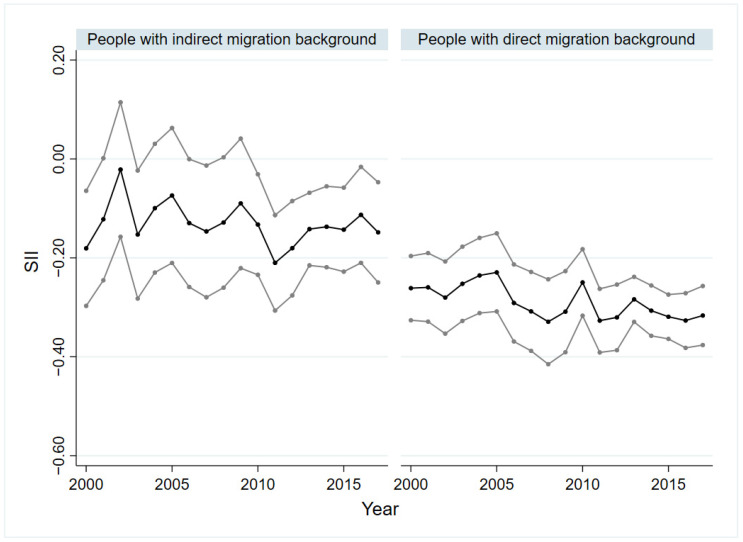
Absolute socio-economic inequalities in SRH in people with direct and indirect migration background, 2000–2017. SRH: self-rated health; SII: slope index of inequality; black line: regression coefficient; grey line: 95% confidence interval.

**Figure 4 ijerph-19-08304-f004:**
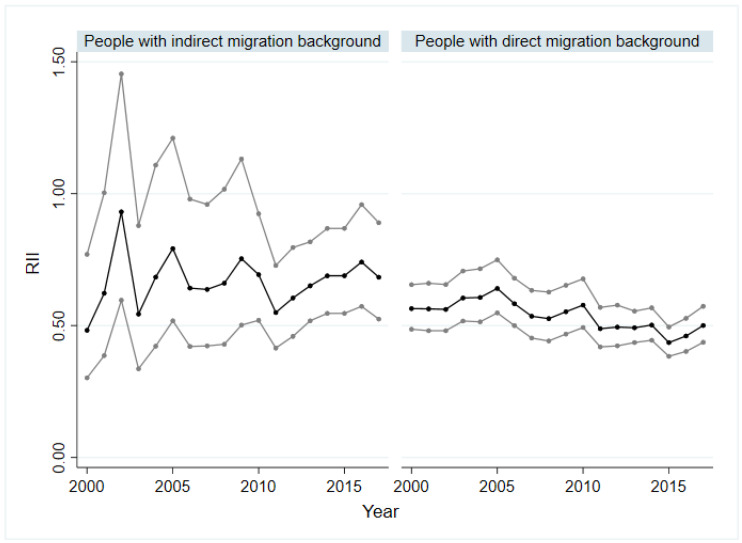
Relative socio-economic inequalities in SRH in people with direct and indirect migration background, 2000–2017. SRH: self-rated health; RII: relative index of inequality; black line: regression coefficient; grey line: 95% confidence interval.

**Table 1 ijerph-19-08304-t001:** Socio-demographic, socio-economic and health-related characteristics of the sample from the Socio-Economic Panel in Germany, 1995–2017, *N* = 492,489 observations.

Year	1995	2000	2005	2010	2015	Total, 1995–2017
**Sex,** * **n** * **(%)**						
Male	6527 (48.2)	11,608 (48)	8980 (47.51)	11,512 (45.54)	11,936 (45.85)	231,857 (46.85)
Female	7015 (51.8)	12,574 (52)	9923 (52.49)	13,765 (54.46)	14,096 (54.15)	259,997 (53.15)
**Age,** * **n** * **(%)**						
18–39 years	6396 (47.23)	9604 (39.72)	6685 (35.36)	9874 (39.06)	9184 (35.28)	183,905 (38.68)
40–65 years	5581 (41.21)	10,919 (45.16)	8579 (45.38)	11,152 (44.12)	12,203 (46.88)	223,314 (44.87)
>65 years	1565 (11.56)	3657 (15.12)	3639 (19.25)	4251 (16.82)	4642 (17.83)	84,616 (16.45)
**Migration background,** * **n** * **(%)**						
Individuals without migration background	10,147 (74.93)	19,985 (82.64)	15,784 (83.50)	20,855 (82.51)	18,226 (70.01)	383,017 (78.75)
Individuals with migration background	3395 (25.07)	4197 (17.36)	3119 (16.50)	4422 (17.49)	7807 (29.99)	108,842 (21.25)
**Migrant status,** * **n** * **(%)**						
Indirect migration background	459 (13.99)	778 (19.41)	739 (24.26)	1244 (28.41)	1835 (23.68)	23,635 (22.51)
Direct migration background	2744 (83.66)	3099 (77.32)	2180 (71.57)	2792 (63.77)	5251 (67.77)	67,711 (71.53)
Asylum seeker and refugee	77 (2.35)	131 (3.27)	127 (4.17)	342 (7.81)	662 (8.54)	15,587 (5.96)
**SES,** * **n** * **(%)**						
Low	4059 (30.52)	6275 (26.52)	3585 (19.89)	4940 (20.45)	4011 (16.18)	100,418 (22.00)
Moderate	7960 (59.85)	14,435 (61.00)	11,455 (63.54)	14,623 (60.52)	15,025 (60.63)	286,499 (61.09)
High	1281 (9.63)	2953 (12.48)	2987 (16.57)	4598 (19.03)	5747 (23.19)	82,434 (16.90)
**Educational level, *n* (%)**						
Low	3117 (23.39)	4320 (18.18)	2844 (15.63)	3287 (13.46)	3696 (14.85)	80,405 (16.50)
Moderate	7452 (55.92)	13,758 (57.91)	10,571 (58.09)	13,545 (55.45)	12,730 (51.13)	259,371 (55.50)
High	2757 (20.69)	5679 (23.90)	4784 (26.29)	7597 (31.10)	8469 (34.02)	133,688 (28.00)
**Income**						
Mean	1122.07	1283.20	1429.05	1487.13	1668.97	1455.00
Standard deviation	606.92	647.52	780.73	860.25	938.27	858.00
**Occupational level, *n* (%)**						
No occupation	2529 (18.71)	6871 (28.52)	3755 (20.06)	6377 (25.52)	5455 (21.06)	114,520 (23.31)
Low	6225 (46.06)	8831 (36.66)	7399 (39.54)	8825 (35.32)	9954 (38.43)	184,796 (38.46)
Moderate	3558 (26.32)	5962 (24.75)	5342 (28.54)	6607 (26.44)	7006 (27.05)	130,198 (26.56)
High	1204 (8.91)	2424 (10.06)	2219 (11.86)	3178 (12.72)	3488 (13.47)	57,956 (11.67)
**SRH, *n* (%)**						
Poor	6862 (50.83)	12,312 (51.00)	9037 (47.98)	13,165 (52.13)	13,315 (51.22)	244,509 (50.77)
Good	6638 (49.17)	11,829 (49.00)	9799 (52.02)	12,088 (47.87)	12,679 (48.78)	239,975 (49.23)
**Total n**	19,947	34,302	30,339	45,977	26,108	849,190

SES: socio-economic status. SRH: self-rated health.

## Data Availability

The data are public-use data and can be obtained via SOEP/DIW upon reasonable request.

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
