# Peer review of "Trends and Changes in Socio-Economic Inequality in Self-Rated Health among Migrants and Non-Migrants: Repeated Cross-Sectional Analysis of National Survey Data in Germany, 1995–2017"

_ijerph, 2022, doi:10.3390/ijerph19148304_

Round 1

Reviewer 1 Report

A very interesting and well-done work focused on quantifying the magnitude of socio-economic inequalities in health among people with direct and indirect migration backgrounds living in Germany. Additionally, it was easy to read.

Abstract. Ok

Introduction. Very well structured and fully justified.

Material and methods. A clear description of the study design and operational definition of variables. The analysis is appropriate to answer the objective.

Results. Well described. The use of figures allows easy visualization of the results over time.

Discussion. Ok

Author Response

Reviewer comments (Round 1)

Reviewer #1: A very interesting and well-done work focused on quantifying the magnitude of socio-economic inequalities in health among people with direct and indirect migration backgrounds living in Germany. Additionally, it was easy to read.

Abstract. Ok

Introduction. Very well structured and fully justified.

Material and methods. A clear description of the study design and operational definition of variables. The analysis is appropriate to answer the objective.

Results. Well described. The use of figures allows easy visualization of the results over time.

Discussion. Ok

R: Thank you for your comments.

Reviewer 2 Report

1. It is not clear what is the Authors main contribution to the existing knowledge and literature. It should be described in Introduction section. 

2. It is not clear why “high income” SOEP-sample was exluded. This needs additional clarification.

3. The meaning of "the slope (SII) and relative index of inequality" should be explained in more detail.

4. I would recommend to describe practical implications (e.g. in terms of healthcare policy) that result from the obtained results.

Author Response

Reviewer comments (Round 1)

Reviewer #2: It is not clear what is the Authors main contribution to the existing knowledge and literature. It should be described in Introduction section. 

R: Thank you. Our contribution to the existing knowledge and literature is the focus on migrant population in Germany. We outlined this in the last paragraph of our introduction (p. 2): In Germany, few studies have been conducted on changes in socio-economic inequalities in health over time [7], and those that exist focus only on the general population. […] the magnitude and the temporal dynamics of socio-economic inequalities in health among migrants remains unclear. We hence aimed to quantify the magnitude of socio-economic inequalities in health among people with direct and indirect migration background living in Germany.

Reviewer #2: It is not clear why “high income” SOEP-sample was excluded. This needs additional clarification.

R: Thank you. We added an explanation for the exclusion of the high-income sample (p. 2): We excluded the so called “high income” SOEP-sample, a sample which consists of extraordinary top-income individuals from Germany, from our analysis to avoid an overrepresentation of high-income groups in our study population, which can lead to biases with regard to the socio-economic and income-specific characteristics examined.

Reviewer #2: The meaning of "the slope (SII) and relative index of inequality" should be explained in more detail.

R: Thank you. We added two sentences for a better understanding of SII and RII (p. 3-4): The SII indicates the absolute difference in health status between the group with the lowest SES and the group with the highest SES, taking into account the health status of all of the groups in-between and the underlying population sizes of each group. The RII can be interpreted according to a relative risk, i.e. the relative difference in health status between the lowest and the highest SES group [24].

Reviewer #2: I would recommend to describe practical implications (e.g. in terms of healthcare policy) that result from the obtained results.

R: Thank you. Because of the limited characters we cannot elaborate on practical implications in detail, but we added a recommendation in our discussion (p. 14): Furthermore, interventions are needed that acknowledge migrants as an heterogenous population group and focus on improving the access to health care of those with a lower SES to overcome the socioeconomic inequities in health.

Reviewer 3 Report

The paper describes a repeated cross sectional study aimed at quantifying socio-economic inequalities in health and assessing temporal trends and changes between 1995-2017 among migrants and non-migrants using data from a large survey in Germany.  This is a very topical subject and an interesting read with a well explained rationale and well planned research. Operationalisation of variables and statistical analysis, although complex, are in general thoroughly described. Results are clearly presented in most tables and figures. However,I would like to make a few minor comments/suggestions        ( this is made slightly more difficult as the manuscript I received has a problem with page numbers and  only has number lines starting at page 7/16 which was marked again 1/16) for the authors to consider:

Materials and Methods 

1.     Knowing the first SOEP survey was in 1984 and that data for the migrants subgroups were only  analysed from 2000 as previously sample sizes were too small, was there a particular reason for the analysis to start in the year 1995 ? If so this could perhaps be explained in the materials and methods section.

2.     At the beginning of the health outcome subsection, self-rated general health is defined as independent variable. Being the outcome, should it be dependent variable ?

3.     It seems like there is no reference to the software used for statistical analysis - this could be added.

Results

4.     To facilitate easier understanding, I would suggest  to add 1995-2017  to Total in 7th column of  table 1 - Total 1995-2017 

5.     In table 1 SRH has 2 categories low and high . I couldn’t find this classification in the definition of variables in the text . What is low – poor ? Perhaps poor and good would make for an easier reading of the table.

6.     I would suggest to add to sentence on line numbered as 8 that missing values are included in table S1 for each of the variables. 

Discussion

7.     In line numbered as 157 it reads “ limitations of subjective information “, I interpret it as self- rated health being subjective, but perhaps the authors  could specify to which information they are referring to.

8.     Line 159 “ According to the variables SRH and SES weaknesses refer to……………” – in  this sentence, “according to “ could be perhaps replaced with “concerning the variables” or “referring to the variables “

Author Response

Reviewer comments (Round 1)

Reviewer #3: Knowing the first SOEP survey was in 1984 and that data for the migrants subgroups were only  analysed from 2000 as previously sample sizes were too small, was there a particular reason for the analysis to start in the year 1995 ? If so this could perhaps be explained in the materials and methods section.

R: Thank you. We added a short sentence in the methods part (p. 2): […] there are subgroup-specific surveys that have been added and expanded since the first survey in 1984. These include, among others, an immigration sample which was added in 1995 […].

Reviewer #3: At the beginning of the health outcome subsection, self-rated general health is defined as independent variable. Being the outcome, should it be dependent variable?

R: Thank you. This was indeed a mistake. We changed the word to “dependent” (p. 3)

Reviewer #3: It seems like there is no reference to the software used for statistical analysis - this could be added.

R: Thank you. We added: Analysis were conducted with Stata Version 16.0. (p. 3)

Reviewer #3: To facilitate easier understanding, I would suggest to add 1995-2017  to Total in 7th column of  table 1 - Total 1995-2017 

R: Thank you. We have added the years following your suggestion.

Reviewer #3: In table 1 SRH has 2 categories low and high. I couldn’t find this classification in the definition of variables in the text. What is low – poor? Perhaps poor and good would make for an easier reading of the table.

R: Thank you. We changed the wording (low to poor; high to good) in table 1 as well as on page 8.

Reviewer #3: I would suggest to add to sentence on line numbered as 8 that missing values are included in table S1 for each of the variables. 

R: Thank you. We added: […] including missing values for each variable described. (p. 8)

Reviewer #3: In line numbered as 157 it reads “limitations of subjective information “, I interpret it as self- rated health being subjective, but perhaps the authors could specify to which information they are referring to.

R: Thank you. We added: as the data did not derive from objective measured information, especially with regard to the SRH […] (p. 14)

Reviewer #3: Line 159 “According to the variables SRH and SES weaknesses refer to……………” – in this sentence, “according to “could be perhaps replaced with “concerning the variables” or “referring to the variables “

R: Thank you. We replaced “according to” with “concerning” (p. 14)

Reference

24 Regidor, E. Measures of health inequalities: part 2. Journal of epidemiology and community health 2004, 58, 900-903, doi:10.1136/jech.2004.023036.